# EXPLOITING CROSS-LINGUAL SUBWORD SIMILARITIES IN LOW-RESOURCE DOCUMENT CLASSIFICATION

## ABSTRACT

Text classification must sometimes be applied in situations with no training data in a target language. However, training data may be available in a *related* language. We introduce a cross-lingual document classification framework (CACO) between related language pairs. To best use limited training data, our transfer learning scheme exploits cross-lingual subword similarity by jointly training a character-based embedder and a word-based classifier. The embedder derives vector representations for input words from their written forms, and the classifier makes predictions based on the word vectors. We use a joint character representation for both the source language and the target language, which allows the embedder to generalize knowledge about source language words to target language words with similar forms. We propose a multi-task objective that can further improve the model if additional cross-lingual or monolingual resources are available. CACO models trained under low-resource settings rival cross-lingual word embedding models trained under high-resource settings on related language pairs.

## 1 INTRODUCTION: CLASSIFIERS ACROSS LANGUAGES

Building an accurate document classifier with traditional supervised learning techniques requires a large labeled corpus. Unfortunately, annotated data is often unavailable (or unreliable) in low-resource languages. Sometimes the need for a low-resource classifier is acute: an earthquake in Haiti,[1] unrest in Ukraine, or food shortages in East Africa. Cross-lingual document classification (Klementiev et al., 2012, CLDC) attacks this problem by using annotated dataset from a *source* language to build classifiers for a *target* language.

The key challenge of CLDC is to find a shared representation for documents from both languages. Once we have a cross-lingual feature space, we can train a classifier on source language documents and apply it on target language documents. Previous work uses a bilingual lexicon (Wu et al., 2008; Shi et al., 2010; Andrade et al., 2015), machine translation (Banea et al., 2008; Wan, 2009; Zhou et al., 2016, MT), topic models (Mimno et al., 2009), or pre-trained cross-lingual word embeddings (Klementiev et al., 2012; Chen et al., 2016, CLWE) to extract cross-lingual features. However, this may be impossible in low-resource languages, as these methods require some combination of large parallel or comparable text, high-coverage dictionaries, and monolingual corpora.

We propose a new CLDC method for a truly low-resource setting, where unlabeled or parallel data in target language is also limited or unavailable. Our system, **C**lassification **A**ided by **C**onvergent **O**rthography (CACO) capitalizes on subword similarities between related language pairs. When the source language and the target language come from the same family, they share cognate words, many of which have similar forms and semantics. For example, "religious" in English, "religieux" in French, and "religioso" in Italian and Spanish share the same meaning.

Previous CLDC methods treat words as atomic symbols and ignore subword patterns. We instead use a bi-level model with two components: a character-based *embedder* and a word-based *classifier*. The embedder creates vectors for input words from their character sequences. Using the word embeddings, the classifier then labels the document. We hope the embedder can learn morpho-semantic regularities, while the classifier connects lexical semantics to labels. To allow cross-lingual transfer, we use a single model for both languages, and we share character embeddings between source and target

---

[1] https://www.ushahidi.com/categories/haiti

languages. We jointly train the embedder and the classifier on annotated source language documents. The embedder transfers knowledge about source language words to target language words with similar orthographic features.

If we have additional unlabeled data such as a dictionary, pre-trained word embeddings, or parallel text, we can fine-tune the model with multi-task learning. Specifically, we encourage the embedder to produce similar word embeddings for dictionary word pairs, which captures patterns between cognate pairs. We also teach the embedder to MIMICK pre-trained word embeddings in the source language (Pinter et al., 2017), which exposes the model to more word types. When we have a good reference model in a high resource language, we can train the model to match the output of the reference model on parallel text.

We experiment on CLDC between nine related language pairs on two datasets. CACO can match the accuracy of CLWE-based models without using any target language data, and fine-tuning the embedder with a small amount of additional resources further improves the accuracy of CACO.

## 2 CLASSIFICATION AIDED BY CONVERGENT ORTHOGRAPHY

This section introduces our model, CACO, which trains a multilingual document classifier using labeled datasets in a source language $\mathcal{S}$ and applies the classifier to a low-resource target language $\mathcal{T}$.

### 2.1 MODEL ARCHITECTURE

Our model has a two-level architecture (Figure 1) that includes a character-level embedder $e$ and a word-level classifier $f$. The embedder $e$ takes characters as inputs and produces a word embedding vector. The classifier then computes a label distribution from the embeddings of all input words. Formally, let $\mathbf{w}$ be an input document with a sequence of tokens $\mathbf{w} = \langle w_1, w_2, \cdots, w_n \rangle$. Our model maps the document $\mathbf{w}$ to a distribution over possible labels $y$ in two steps. First, we generate a word embedding $\mathbf{v}_i$ for each input word $w_i$ using the embedder $e$:

$$\mathbf{v}_i = e(w_i). \tag{1}$$

We then feed the word embeddings to the classifier $f$ to compute the distribution over labels $y$:

$$p(y \mid \mathbf{w}) = f(\langle \mathbf{v}_1, \mathbf{v}_2, \cdots, \mathbf{v}_n \rangle). \tag{2}$$

We can use any sequence model for the embedder $e$ and the classifier $f$. For our experiments, we use a bidirectional LSTM (Graves & Schmidhuber, 2005, BI-LSTM) embedder and a deep averaging network (Iyyer et al., 2015, DAN) classifier.

**BI-LSTM Embedder**: BI-LSTM is a powerful sequence model that captures complex non-local dependencies. Character-level BI-LSTM embedders are successfully used in many natural language processing tasks (Ling et al., 2015a;b; Ballesteros et al., 2015; Lample et al., 2016). To embed a word $w$, we pass its character sequence $\mathbf{c}$ to a left-to-right LSTM and the reversed character sequence $\mathbf{c}'$ to a right-to-left LSTM. We concatenate the final hidden states of the two LSTM and apply a linear transformation:

$$\mathbf{e}(w) = \mathbf{W}_e \cdot [\overrightarrow{\text{LSTM}}(\mathbf{c}); \overleftarrow{\text{LSTM}}(\mathbf{c}')] + \mathbf{b}_e, \tag{3}$$

where the functions $\overrightarrow{\text{LSTM}}$ and $\overleftarrow{\text{LSTM}}$ compute the final hidden states of the two LSTMs.

**DAN Classifier**: A DAN is an unordered model that passes the arithmetic mean of the input word embeddings through a multilayer perceptron and feed the final layer's representation to a softmax layer. We choose DAN because it ignores cross-lingual variations in word orders (i.e., syntax) and thus generalizes well in CLDC. Despite its simplicity, DAN achieves state-of-the-art accuracies on both monolingual and cross-lingual document classification (Iyyer et al., 2015; Chen et al., 2016).

Let $\mathbf{v}_1, \mathbf{v}_2, \cdots, \mathbf{v}_n$ be the input word embeddings. A DAN uses the average of the word embeddings as the document representation $\mathbf{z}_0$:

$$\mathbf{z}_0 = \text{mean}(\mathbf{v}_1, \mathbf{v}_2, \cdots, \mathbf{v}_n), \tag{4}$$

and $\mathbf{z}_0$ is passed through $k$ layers of non-linearity:

$$\mathbf{z}_i = g(\mathbf{W}_i \cdot \mathbf{z}_{i-1} + \mathbf{b}_i), \tag{5}$$

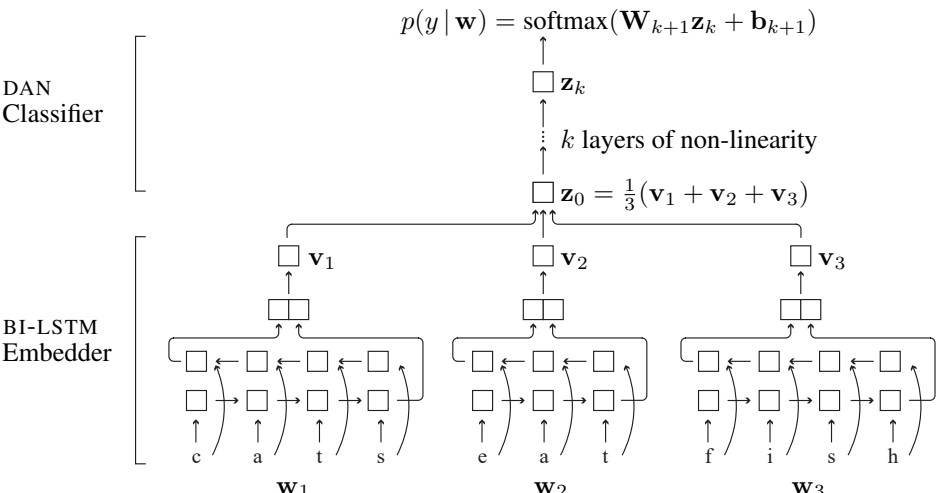

Figure 1: In CACO, each input word $\mathbf{w}_i$ is mapped to a word vector $\mathbf{v}_i$ by passing its characters through a BI-LSTM embedder. The average of the word vectors $\mathbf{z}_0$ is then passed through $k$ layers of non-linearity and a final softmax layer to predict the label $y$.

where $i$ ranges from 1 to $k$, and $g$ is a non-linear activation function. The final representation $\mathbf{z}_k$ is passed to a softmax layer to obtain a distribution over the label $y$,

$$p(y \mid \mathbf{w}) = \text{softmax}(\mathbf{W}_{k+1}\mathbf{z}_k + \mathbf{b}_{k+1}). \tag{6}$$

## 2.2 CHARACTER-LEVEL CROSS-LINGUAL TRANSFER

To transfer character-level information across languages, the embedder uses the same character embeddings for both languages. The character-level BI-LSTM vocabulary is the union of the alphabets for the two languages, and the embedder does not differentiate identical characters from different languages. For example, a Spanish "a" has the same character embedding as a French "a". Consequently, the embedder maps words with similar forms from both languages to similar vectors.

If the source language and the target language are lexically similar, the embedder can generalize knowledge learned about source language words to target language words through shared orthographic features. As an example, if the model learns that the Spanish word "religioso" is predictive of label $y$, the model automatically infers that "religioso" in Italian is also predictive of $y$, even though the model never sees any Italian text.

## 2.3 OBJECTIVE

Our main training objective is supervised document classification. We jointly train the classifier and the embedder to minimize average negative log-likelihood on labeled source language documents $S$:

$$L_s(\theta) = -\frac{1}{|S|} \sum_{\langle \mathbf{w}, y \rangle \in S} \log p(y \mid \mathbf{w}; \theta), \tag{7}$$

where $\theta$ is model parameter, and $S$ contains source language examples with words $\mathbf{w}$ and label $y$.

Sometimes we have additional resources for the source or target language. We use them to improve our model with multi-task learning (Collobert et al., 2011) via three auxiliary tasks.

**Word Translation (DICT)**: There are many patterns when translating cognate words between related languages. For example, Italian "e" often becomes "ie" when translating into Spanish. "Tempo" (time) in Italian becomes "tiempo" in Spanish, and "concerto" (concert) in Italian becomes "concierto" in Spanish. The embedder can learn these word translation patterns from a bilingual dictionary.

Let $D$ be a bilingual dictionary with a set of word pairs $\langle w, w' \rangle$, where $w$ and $w'$ are translations of each other. We add a term to our objective to minimize average squared Euclidean distances between

the embeddings of translation pairs (Mikolov et al., 2013):

$$L_d(\theta) = \frac{1}{|D|} \sum_{\langle w, w' \rangle \in D} \|\mathbf{e}(w) - \mathbf{e}(w')\|_2^2. \tag{8}$$

A bilingual dictionary can also reveal false friends—words from different languages that have similar forms but different meanings. For example, the Italian word "burro" (butter) is "mantequilla" in Spanish, while the Spanish word "burro" (donkey) is "asino" in Italian. However, our current model's joint character embedding space treats "burro" (Spanish) and "burro" (Italian) identically. Therefore, we consider a variant of CACO with **language identifiers** (in § 4 these variants are denoted by $^+$). Specifically, the vocabulary of the embedder now includes two special language identifier characters for the two languages. We prepend and append the identifier to each input word. This allows the embedder to differentiate words from different languages, but the added expressiveness might induce overfitting to the dictionary and prevent generalization through orthographic features.

**Mimicking Word Embeddings (MIM)**: Monolingual text classifiers often benefit from initializing embeddings with word vectors pre-trained on large unlabeled corpus (Collobert et al., 2011). This semi-supervised learning strategy helps the model generalize to word types outside labeled training data. Similarly, our embedder can MIMICK (Pinter et al., 2017; Stratos, 2017) an existing *source language* word embeddings to learn and transfer useful representations.

Suppose we have a pre-trained source language word embedding matrix $\mathbf{X}$ with $V$ rows. The $i$-th row $\mathbf{x}_i$ is a vector for the $i$-th word type $w_i$. We add an objective to minimize the average squared Euclidean distances between the output of the embedder and $\mathbf{X}$:

$$L_e(\theta) = \frac{1}{V} \sum_{i=1}^{V} \|\mathbf{e}(w_i) - \mathbf{x}_i\|_2^2. \tag{9}$$

**Parallel Projection**: Sometimes we have a reliable reference classifier in another high-resource language $\mathcal{S}'$ (for example, English). If we have parallel text between $\mathcal{S}$ and $\mathcal{S}'$, we can use parallel projection (Yarowsky et al., 2001) to add additional training signal. Let $P$ be a set of parallel documents $\langle \mathbf{w}, \mathbf{w}' \rangle$, where $\mathbf{w}$ is from source language $\mathcal{S}$, and $\mathbf{w}'$ is the translation of $\mathbf{w}$ in $\mathcal{S}'$. We add another objective term to minimize the average Kullback-Leibler divergence between the predictions of our model predictions and the reference model:

$$L_p(\theta) = \frac{1}{|P|} \sum_{\langle \mathbf{w}, \mathbf{w}' \rangle \in P} D_{KL}[p'(\cdot \mid \mathbf{w}') \parallel p(\cdot \mid \mathbf{w})], \tag{10}$$

where $p'$ is the output of the reference classifier (in language $\mathcal{S}'$), and $p$ is the output of CACO. In § 4, we mark models that use parallel projection with a superscript "P".

We train our model on the four tasks jointly. Our final objective is:

$$L(\theta) = \underbrace{L_s(\theta)}_{\text{classifier}} + \lambda_d \underbrace{L_d(\theta)}_{\text{dictionary}} + \lambda_e \underbrace{L_e(\theta)}_{\text{embed}} + \lambda_p \underbrace{L_p(\theta)}_{\text{parallel}}, \tag{11}$$

where the hyperparameters $\lambda_d$, $\lambda_e$, and $\lambda_p$ trade off between the four tasks.

## 3 RELATED WORK

Previous CLDC methods are typically word-based and rely on one of the following cross-lingual signals to transfer knowledge: large bilingual lexicons (Wu et al., 2008; Shi et al., 2010; Andrade et al., 2015), MT systems (Banea et al., 2008; Wan, 2009; Zhou et al., 2016), or cross-lingual word representations (Klementiev et al., 2012; Chen et al., 2016). Unfortunately, these resources are not available in every language. Recent work proposes unsupervised methods for building CLWE (Artetxe et al., 2018a; Conneau et al., 2018; Zhang et al., 2017a;b) and MT systems (Artetxe et al., 2018b; Lample et al., 2018a;b) without any cross-lingual signal. However, these methods still require large monolingual corpora in the target language, and they might fail when the monolingual corpora for the two languages come from different domains (Søgaard et al., 2018). In contrast, CACO is much more

Table 1: CLDC experiments between eight related European language pairs on RCV2 topic identification. The CACO models are competitive with DAN models that use **far more resources**. The combined model (COM) achieves the highest average test accuracy. We **boldface** the best result for each row, underline CACO results that outperform at least one DAN model.

| | | DAN | | | CACO | | | | | | |
|---|---|---|---|---|---|---|---|---|---|---|---|
| source | target | mCCA | mClu | SUP | SRC | DICT | DICT$^+$ | MIM | ALL | ALL$^+$ | COM |
| DA | SV | 69.3 | 63.0 | 59.7 | 56.0 | 62.8 | 62.8 | 60.4 | 62.9 | 62.9 | **69.7** |
| SV | DA | 51.4 | 40.8 | 54.7 | 56.7 | 60.2 | 63.4 | 58.4 | 62.2 | 56.0 | **67.5** |
| FR | ES | 63.9 | **71.8** | 56.6 | 49.6 | 59.3 | 59.8 | 48.3 | 57.4 | 60.2 | 70.8 |
| IT | ES | 43.4 | 55.6 | 56.6 | 50.2 | 54.6 | 48.5 | 51.4 | 54.7 | 44.9 | **63.5** |
| ES | FR | **63.1** | 60.0 | 48.9 | 48.5 | 49.7 | 43.2 | 49.2 | 48.8 | 47.4 | 61.3 |
| IT | FR | 26.7 | **66.5** | 48.9 | 45.9 | 52.1 | 52.2 | 46.6 | 48.2 | 47.2 | 62.8 |
| FR | IT | 43.6 | 59.6 | 44.9 | 43.3 | 53.2 | 54.4 | 44.3 | 51.2 | 48.5 | **60.2** |
| ES | IT | 51.3 | 51.4 | 44.9 | 49.7 | 53.5 | 47.6 | 53.4 | 52.5 | 48.2 | **59.7** |
| | average | 51.6 | 58.6 | 51.9 | 50.0 | 55.7 | 54.0 | 51.5 | 54.7 | 51.9 | **64.5** |

Table 2: CLDC experiments between Amharic and Tigrinya on LORELEI disaster response dataset. CACO models (bottom) outperform mClu models (top) without using any target language data. For AM-TI, parallel projection (SRC$^P$ and MIM$^P$) further improves CACO models. We do not experiment with parallel projection on Tigrinya because we cannot find enough unlabeled parallel text in the language pack. We **boldface** the best result for each row and underline CACO results that outperform one DAN model.

| | | DAN | | CACO | | | |
|---|---|---|---|---|---|---|---|
| source | target | mCCA | mClu | SRC | MIM | SRC$^P$ | MIM$^P$ |
| AM | TI | **59.1** | 55.8 | 55.5 | 56.3 | 57.0 | 57.6 |
| TI | AM | **58.1** | 50.0 | 56.8 | 55.1 | - | - |

data-efficient. By exploiting character-level similarities between related languages, CACO trained with few or no target language data is competitive with CLWE-based models.

Our work builds on the success of character-level BI-LSTM embedders in monolingual NLP tasks, including language modeling and part-of-speech tagging (Ling et al., 2015a), named entity recognition (Lample et al., 2016), and dependency parsing (Ballesteros et al., 2015). Character-level embedders can generate useful representations for rare and unseen words, especially for morphologically rich languages. We extend this to CLDC. Recent work also uses character BI-LSTMs to directly reconstruct pre-trained monolingual word embeddings (Pinter et al., 2017; Stratos, 2017), which motivates our MIMICK objective.

Cross-lingual transfer at the character-level is successfully used in low-resource paradigm completion (Kann et al., 2017), morphological tagging (Cotterell & Heigold, 2017), POS tagging (Kim et al., 2017), and named entity recognition (Cotterell & Duh, 2017; Lin et al., 2018), where the authors train a character-level model jointly on a small labeled corpus in target language and a large labeled corpus in source language. Our method is similar in spirit, but we focus on CLDC, where it is less obvious if orthographic features are helpful. Moreover, we introduce a novel multi-task objective to use different types of monolingual and cross-lingual resources; we can match CLWE-based models using little or no target language data.

## 4 EXPERIMENTS

We evaluate our method on two CLDC datasets: Reuters multilingual corpora (Lewis et al., 2004, RCV2) and LORELEI language packs (Strassel & Tracey, 2016). To show the effectiveness of our method, we train CACO models in a low-resource setting and compare two CLWE-based model trained in a high-resource setting and a supervised monolingual model. **Our goal is not to beat the**

Table 3: Results of CLDC experiments using two source languages. Models trained on two source languages are generally better than models trained on only one source language (Table 1). We **boldface** the best result for each row, underline CACO results that outperform at least one DAN model.

| | | DAN | | CACO | | | | | |
|---|---|---|---|---|---|---|---|---|---|
| source | target | mCCA | mClu | SRC | DICT | DICT$^+$ | MIM | ALL | ALL$^+$ |
| 🇫🇷🇮🇹 FR+IT | 🇪🇸 ES | 74.2 | **77.0** | 58.8 | 67.0 | 61.1 | 55.8 | 65.3 | 60.4 |
| 🇪🇸🇮🇹 ES+IT | 🇫🇷 FR | 55.2 | **65.3** | 51.8 | 55.8 | 50.3 | 50.3 | 56.0 | 52.5 |
| 🇪🇸🇫🇷 ES+FR | 🇮🇹 IT | 45.4 | **61.0** | 53.2 | 56.1 | 55.6 | 55.9 | 56.5 | 53.2 |
| | average | 58.3 | **67.8** | 54.6 | 59.6 | 55.7 | 54.0 | 59.3 | 55.4 |

Table 4: Word translation accuracies (P@1) for different embeddings. The CACO embeddings are generated by the embedder of a SRC model trained on the source language. Without any cross-lingual signal, the CACO embedder has similar word translation accuracy as mCCA and mCluster, which are supervisedly trained on large dictionaries.

| source | target | mCCA | mClu | CACO |
|---|---|---|---|---|
| 🇪🇸 ES | 🇫🇷 FR | 36.8 | 40.8 | 31.1 |
| 🇪🇸 ES | 🇮🇹 IT | 44.0 | 35.7 | 33.1 |
| 🇫🇷 FR | 🇪🇸 ES | 34.0 | 40.2 | 30.9 |
| 🇫🇷 FR | 🇮🇹 IT | 33.5 | 40.3 | 29.6 |
| 🇮🇹 IT | 🇪🇸 ES | 42.1 | 37.8 | 37.5 |
| 🇮🇹 IT | 🇫🇷 FR | 35.6 | 39.8 | 36.4 |
| | average | 37.7 | 39.1 | 33.1 |

**baselines, but to show that CACO is competitive with the baselines despite training on much less target language data.**

## 4.1 CLASSIFICATION DATASET

Our first dataset is RCV2, a multilingual collection of news stories labeled with topics (Lewis et al., 2004).[2] Following Klementiev et al. (2012), we remove documents with multiple topic labels. For each language, we sample 1,500 training documents and 200 test documents with balanced labels. We conduct CLDC experiments between two North Germanic languages, Danish (DA) and Swedish (SV), and three Romance languages, French (FR), Italian (IT), and Spanish (ES).

To test CACO on truly low-resource languages, we build a second CLDC dataset with famine-related documents sampled from Tigrinya (TI) and Amharic (AM) LORELEI language packs (Strassel & Tracey, 2016).[3] We train models to classify whether the document describes widespread crime or not.[4] The Amharic language pack does not have annotations, so we manually label Amharic sentences based on English reference translations.[5] Our final dataset contains 394 Tigrinya and 370 Amharic documents with balanced labels.

## 4.2 MODELS

We experiment with several variants of CACO that use different resources. The **SRC** model is only trained on labeled source language documents and do not use any unlabeled data. The **DICT** model is trained on both labeled documents and parallel dictionary with the supervised objective and the word translation objective. The **MIM** model is jointly trained to label source language documents and

---

[2]Corporate/Industrial (CCAT), Economics (ECAT), Government/Social (GCAT), and Markets (MCAT).

[3]Amharic: LDC2016E87. Tigrinya: LDC2017E57.

[4]For Tigrinya documents, the labels are extracted from the situation frame annotation in the language pack. We mark all documents with a "crimeviolence" situation frame as positive.

[5]Annotations available after blind review.

mimick the source language part of multiCCA embeddings. The **ALL** model is trained using both the word translation and the mimick tasks.

All of the above models do not use language identifier characters and therefore cannot distinguish words from different languages with identical forms. For DICT and ALL, we also experiment with their variants that prepend and append a language identifier character to each word. We name these models **DICT⁺** and **ALL⁺**. We also use parallel projection to provide more classification signals for some models. We mark these models with a superscript "P".

For comparison, we train two word-based DANs with pre-trained CLWE features. We use the multiCCA (**mCCA**) and multiCluster (**mClu**) embeddings (Ammar et al., 2016), which are trained on large corpora with millions of tokens and high-coverage dictionaries with hundreds of thousands of word types. The CLWE-based DANs are strong high-resource models that require large corpora and dictionary for the target language. In contrast, CACO models are trained with few or no target language data in a simulated low-resource setting. **Therefore, it is unfair to directly compare the results of CACO and CLWE-based models**. However, CACO models are often competitive with CLWE-based DANs in our experiment, demonstrating the effectiveness of our proposed method. We also compare CACO with a lightly-supervised monolingual model **SUP**,[6] a DAN trained on fifty labeled target language documents. Without using any target language supervision, CACO models achieves similar (and sometimes higher) test accuracies as SUP.

Finally, we experiment with a model that combines CACO and CLWE. Specifically, we use pre-trained multiCluster CLWE as additional features for the classifier of a SRC model. The combined model on average significantly outperforms both CACO models and clwe-based DAN, which shows that our method is useful even when we have a high-quality cross-lingual word embedding.

### 4.3 DICTIONARY AND PARALLEL TEXT

Some of the CACO models use a dictionary to learn word translation patterns. We train them on the same training dictionary used for learning multiCCA and multiCluster. To simulate the low-resource setting, we sample **only 100 translation pairs** from the original dictionary for CACO.

The original dictionary contains many word types with identical forms. We remove these pairs when sampling the dictionary for DICT and ALL, because without the language identifiers, the embedder always maps words with the same form to the same vector. However, our pilot experiment shows that removing these pairs could hurt the test accuracy for DICT⁺ and ALL⁺. The embedder likely overfits the translation pairs by aggressively tuning the character embedder for the language identifiers. As a result, the embedder behaves very differently for the two languages, which prevents cross-lingual knowledge transfer. Therefore, we keep translation pairs with same surface forms when sampling dictionaries for DICT⁺ and ALL⁺.

The Amharic labeled dataset is very small compared to other languages,[7] so we experiment with the parallel projection technique for the Amharic to Tigrinya CLDC experiment using English-Amharic parallel text. We first train a reference English DAN on a large collection of labeled English documents compiled from other LORELEI language packs. We then use the parallel projection objective to train the CACO models to match the output of the English model on 1,200 English-Amharic parallel documents sampled from the Amharic language pack.[8] We do not use parallel projections on other language pairs, because we have enough labeled examples for the RCV2 languages, and we do not have enough unlabeled parallel text in the Tigrinya language pack.

### 4.4 RESULTS AND ANALYSIS

We train each model using ten different random seeds and report their average test accuracy.[9] We describe training details and hyperparameters in the appendix. Table 1 and Table 2 show average test accuracies of different models on RCV2 and LORELEI on nine related language pairs.

---

[6]We only apply this baseline to RCV2, because the test sets in LORELEI are too small to split further.

[7]Each Amharic document only contains one sentence.

[8]To avoid introducing extra bias, we sample the parallel documents such that the English model output approximately follows a uniform distribution.

[9]For models that use dictionaries, we also re-sample the training dictionary for each run.

CACO vs. DAN: The low-resource CACO models have similar average test accurcy as the high-resource DAN models. The SRC variant does not use any target language data, and yet it outperforms mCCA on three language pairs. When we already have a good CLWE, we can get the best of both world by combining them (COM), which achieves the highest average test accuracy.

**Multi-Task Learning**: Training CACO with multi-task learning further improves the accuracy. For almost all language pairs, the multi-task CACO variants have higher test accuracies than SRC. On RCV2, word translation (DICT) is particularly effective, even with only 100 translation pairs. Interestingly, word translation and mimick tasks together (ALL) do not consistently increase the accuracy over only using the dictionary (DICT). On the LORELEI dataset where labeled document is limited, parallel projection task (SRC$^P$ and MIM$^P$) also significantly increases the accuracies.

**Language Identifier**: The variants that use language identifiers (DICT$^+$ and ALL$^+$) have lower average test accuracies than their counterparts (DICT and ALL). As discussed in §2.3, language identifiers could cause the embedders to overfit the training dictionary. The embedder might learn to behave very differently for the two languages, which prevents generalization across languages.

**Language Relatedness**: We expect CACO to be less effective when the source language and the target language are less close to each other. For comparison, we experiment on RCV2 with transferring between more distantly related language pairs: a North Germanic language and a Romance language. We include the results in the appendix (Table 5). Indeed, CACO models score consistently lower than the DAN models when transferring from a North Germanic source language to a Romance target language. However, CACO models are surprisingly competitive with DAN models when transferring from a Romance language to a North Germanic language, which shows that our method is useful even for loosely related languages. This asymmetry is likely due to morphology differences between the two families.

**Multi-Source Transfer**: We experiment with multi-source transfer learning on RCV2 by training models on *two* Romance languages and testing on another Romance language. Languages can be similar along different dimensions, and therefore adding more source languages may be beneficial. Moreover, using multiple source languages has a regularization effect and prevents the model from overfitting to a single language. For fair comparison, we sample 750 training documents from each source language, so that the multi-source models are still trained on 1,500 training documents (same as the single-source models). We use a similar strategy to sample the training dictionaries and pre-trained word embeddings. Multi-source models (Table 3) consistently outperform single-source models (Table 1) for both CACO and DAN.

**Learned Word Representation**: We evaluate the word representations learned by CACO with a word translation task between French, Italian, and Spanish. Specifically, we use the SRC embedder to generate embeddings for all French, Italian, and Spanish words that appear in both multiCCA and multiCluster vocabulary.[10] Table 4 shows the word translation accuracy (P@1) on the test dictionaries from the MUSE library (Conneau et al., 2018). Although the SRC embedder is not exposed to any cross-lingual signal, it still rivals CLWE on the word translation task by exploiting subword similarities between languages.

**Qualitative Analysis**: To understand how cross-lingual subword similarity can help document classification, we manually compare the output of a DAN model and a CACO model. Specifically, we use the mCCA model and the DICT model from the Spanish to Italian CLDC experiment, and we inspect their predictions on an Italian dev set. Sometimes CACO can avoid the mistakes of DAN by correctly aligning word pairs that are misaligned in the multiCCA embedding space. For example, in multiCCA, "relevancia" (relevance) is the closest Spanish word for the Italian word "interesse" (interest), while the CACO embedder maps both the Italian word "interesse" (interest) and the Spanish word "interesse" (interest) to the same point. Consequently, CACO correctly classifies an Italian document about the interest rate with GCAT (government), while the DAN model predicts MCAT (market). As another example, in multiCCA, the closest Spanish word for the Italian word "anticipazioni" (advance) is "anuncios" (advertisement), and DAN assigns a wrong label to an Italian document about wage advance. The CACO embedder aligns the Italian word "anticipazioni" (advance) with the Spanish word "aprovisionada" (provision) and therefore correctly labels the document.

---

[10]ES: 220,063 words, FR: 219,635 words, IT: 234,366 words.

## 5    CONCLUSION AND FUTURE WORK

We introduce a CLDC framework, CACO, that trains a document classifier with labeled training documents from a related language and optionally a small dictionary, pre-trained embeddings, or parallel text. Our method exploits subword similarities between related languages to generalize from source language data. This technique is particularly useful in low-resource settings where unlabeled monolingual or parallel texts are limited. We provide empirical evaluation of CACO on multiple related language pairs, and CACO matches high resource CLWE-based methods without using monolingual or parallel text from the target language. In the future, we plan to extend our method to related languages with different scripts by combining with a transliteration tool or a grapheme-to-phoneme transducer (Mortensen et al., 2018). We also want to investigate alternative architectures for the embedder, including morpheme-based models (Luong et al., 2013).

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

Table 5: CLDC experiments between languages from different families on RCV2. When transferring from a North Germanic language to a Romance language, CACO models score much lower than DAN models (top). Surprisingly, CACO models are on par with DAN when transferring from a Romance language to a North Germanic language (bottom). We **boldface** the best result for each row, underline CACO results that outperform one DAN model, and double underline CACO results that outperform both DAN models.

| | | DAN | | CACO | | | | | |
|---|---|---|---|---|---|---|---|---|---|
| source | target | mCCA | mClu | SRC | DICT | DICT⁺ | MIM | ALL | ALL⁺ |
| DA | ES | **65.7** | 45.3 | 32.5 | 34.8 | 36.7 | 30.6 | 38.2 | 36.8 |
| DA | FR | 45.9 | **57.2** | 34.1 | 41.8 | 39.9 | 35.5 | 43.3 | 41.3 |
| DA | IT | 47.4 | **52.1** | 36.8 | 43.7 | 36.3 | 37.2 | 41.5 | 37.8 |
| SV | ES | **48.5** | 44.2 | 35.2 | 42.5 | 35.2 | 34.6 | 36.8 | 35.2 |
| SV | FR | **49.0** | 27.3 | 27.4 | 29.9 | 29.0 | 29.1 | 28.3 | 29.3 |
| SV | IT | 40.4 | **49.3** | 34.6 | 36.4 | 35.1 | 33.3 | 35.2 | 35.0 |
| | average | **49.5** | 45.9 | 33.4 | 38.2 | 35.4 | 33.4 | 37.2 | 35.9 |
| ES | DA | **56.7** | 53.4 | 47.7 | 48.3 | 42.9 | 46.1 | 52.0 | 48.0 |
| ES | SV | 52.4 | 53.0 | 50.6 | 53.7 | 39.5 | 48.5 | 51.4 | 42.1 |
| FR | DA | 45.3 | 29.4 | 46.7 | 44.2 | 47.6 | 44.7 | **48.6** | 46.6 |
| FR | SV | **57.2** | 35.4 | 52.9 | 53.2 | 52.8 | 53.6 | 52.8 | 49.6 |
| IT | DA | **48.2** | 45.3 | 36.6 | 43.6 | 40.1 | 34.8 | 43.0 | 43.6 |
| IT | SV | 31.1 | **53.8** | 37.8 | 45.3 | 39.2 | 30.7 | 43.9 | 38.9 |
| | average | 48.5 | 45.1 | 45.4 | 48.1 | 43.7 | 43.1 | **48.6** | 44.8 |

# 6 APPENDIX

**Training details:** For the CLWE-based models, we use forty dimensional multiCCA and multiCluster word embeddings. We use three ReLU layers with 100 hidden units and 0.1 dropout for the CLWE-based DAN models and the DAN classifier of the CACO models. The BI-LSTM embedder uses ten dimensional character embeddings and forty hidden states with no dropout. The outputs of the embedder are forty dimensional word embeddings. We set $\lambda_d$ to 1, $\lambda_e$ to 0.001, and $\lambda_p$ to 1 in the multi-task objective. The hyperparameters are tuned in a pilot Italian-Spanish CLDC experiment using held-out datasets.

All models are trained with ADAM (Kingma & Ba, 2015) with default settings. We run the optimizer for a hundred epochs with mini-batches of sixteen documents. For models that use additional resources, we also sample sixteen examples from each type of training data (translation pairs, pre-trained embeddings, or parallel text) to estimate the gradients of the auxiliary task objectives $L_d$, $L_e$, and $L_p$ (defined in §2.3) at each iteration.

**More RCV2 results:** Table 5 shows the CLDC results on RCV2 between languages from different families, which are discussed in §4.4.

