# OpenReview forum: "Exploiting Cross-Lingual Subword Similarities in Low-Resource Document Classification"
_ICLR.cc/2019/Conference_

### Official Review · AnonReviewer1 · 2018-11-01
**Well-presented but some shortcomings in experiments**

**Rating:** 6
**Confidence:** 3

**Review:**

Overview:

This paper proposes an approach to document classification in a low-resource language using transfer learning from a related higher-resource language. For the case where limited resources are available in the target low-resource language (e.g. a dictionary, pretrained embeddings, parallel text), multi-task learning is incorporated into the model. The approach is evaluated in terms of document classification performance using several combinations of source and target language.

Main strengths:

1. The paper is well written. The model description in Section 2 is very clear and precise.
2. The proposed approach is simple but still shows good performance compared to models trained on corpora and dictionaries in the target language.
3. A large number of empirical experiments are performed to analyse different aspects and the benefits of different target-language resources for multi-task learning.

Main weaknesses:

1. The application of this model to document classification seems to be new (I am not a direct expert in document classification), but the model itself and the components are not (sequence models, transfer learning and multitask learning are well-established). So this raises a concern about novelty (although the experimental results are new).

2. With regards to the experiments, it is stated repeatedly that the DAN model which are compared to uses "far more resources." The best ALL-CACO model also relies on several annotated but "smaller" resources (dictionaries, parallel text, embeddings). Would it be possible to have a baseline where a target-language model is trained on only a small amount of annotated in-domain document classification data in the target language? I am proposing this baseline in order to answer two questions. (i) Given a small amount of in-domain data for the task at hand, how much benefit do we get from additionally using data from a related language? (ii) How much benefit do we get from using target-language resources that do not address the task directly (dictionaries, embeddings) compared with using a "similar" amount of data from the specific task?

Overall feedback:

This is a well-written paper, but I think since the core of the paper lies in its empirical evaluation, the above experiments (or something similar) would greatly strengthen the work.

Edit: I am changing my rating from 5 to 6 based on the authors' response.

---

> ### Author Response · Authors · 2018-11-26
> **Response**
>
> Thank you for your review!
>
> Reviewer 1 mentions that the model components are not new. It is true that our model is built on existing models. However, the novelty of our work lies in the combination of these techniques and the application to cross-lingual document classification.
>
> Following Reviewer 1’s suggestion, we add a baseline that is trained on a small set of 50 labeled documents in the target language. In general, our CACO models perform on par with this lightly-supervised target language model (SUP in Table 1 in our new draft). We hope this further demonstrates the effectiveness of our method. We cannot apply this baseline on the LORELEI dataset since it is too small to split further.

---

### Official Review · AnonReviewer2 · 2018-11-03
**Need more insights**

**Rating:** 6
**Confidence:** 4

**Review:**

Summary: The authors address the task of cross language document classification when there is no training data available in the target language but data is available a closely related language. The authors propose forming character-based embeddings of words to make use of sub-word similarities in closely-related languages. The authors do an extensive evaluation using various combinations of related languages and show improved performance. In particular,  the performance is shown to be competitive with word-based models, which are tied to a requirement of resources involving the original language (such as MT systems, bilingual lexicons, etc). The authors show that their results are boosted when some additional resources (such as bilingual dictionaries of minimal size) are used in a multi-task learning setup.


- I would have liked to see some comparison where your model also uses all the resources available to CLWE based models (for example, larger dictionary, larger parallel corpus, etc)

- It is mentioned that you used parallel projection only for Amharic as for other languages you had enough RCV2 training data. However, it would be interesting to see if you still use parallel projection on top of this.

- I do not completely agree with the statement that CACO models are "not far behind" DAN models. IN Table 1, for most languages the difference is quite high. I understand that your model uses fewer resources but can it bridge the gap by using more resources? Is the model capable of doing so ?

- How did you tune the lambdas in Eqn 11? Any interesting insights from the values of these lambdas? Do these lambda values vary significantly across languages ?

- The argument about why the performance drops when you use language identifiers is not very convincing. Can you please elaborate on this ?

- Why would the performance be better in one directions as compared to another (North Germanic to Romance v/s ROmance to North Germanic). Some explanation is needed here.

- One recurring grievance that I have is that there are no insights/explanations for any results. Why are the gains better for some language pairs? Why is there asymmetry in the results w.r.t direction of transfer ? In what way do 2 languages help as compared to single source language? What is you use more that 2 source languages?

---

> ### Author Response · Authors · 2018-11-26
> **Response**
>
> Thank you for your review!
>
> Reviewer 2 asks how our model performs when using the same resources as the CLWE-based models. In our preliminary experiment, using a larger dictionary slightly improves test accuracy for the CACO models, but the improve becomes marginal as the dictionary is larger. Note that CLWE-based models are strictly more expressive than character-based models and have many more parameters (CLWE learns a separate vector for each word). Therefore, CLWE-based models are more suitable for high-resource settings, while our model is specifically designed for the low-resource setting, which we focus on in this paper.
>
> Our model is still useful in the high-resource setting though. In our updated draft, we experiment on feeding CLWE as extra features to a CACO classifier, and the test accuracies are significantly higher (on average) than only using CLWE as features (COM in Table 1). Therefore our model is useful even when we have enough resources to train a good CLWE.
>
> Reviewer 2 asks what happens if we apply parallel projection on RCV2. In our preliminary experiment, using parallel project does not improve test accuracy on RCV2, because we already have a rather large set of high-quality labeled data. Therefore, we choose not to apply PP on RCV2.
>
> Reviewer 2 asks about how the lambdas in Eq. 11 are tuned. As mentioned in the appendix, we use the same hyperparameters (including the lambdas) for all language pairs, and they are tuned on a held-out set of one “dev” language pair (it-es). In particular, we find it helpful to use a smaller lambda_e, which implies that the mimick task is less helpful than the other two auxiliary tasks.
>
> Reviewer 2 asks why language identifiers hurts the performance. Using language identifier allows the embedder to behave differently for the two languages. In practice, this added expressiveness could lead to overfitting the training dictionary. Consequently, the embedder might assign very different representations to orthographically similar words from the two languages. This could prevent generalization through orthographic features and decrease test accuracy.
>
> Reviewer 2 asks why our experiment results are asymmetric, and why the performance gains are better for some language pairs. In general, the effectiveness of *any* existing cross-lingual transfer technique varies across languages, and there is no guarantee that the results should be symmetric. In our case, we hypothesize the differences in morphology between languages plays an important role. This is an important research question that we wish to investigate in future work.
>
> Reviewer 2 asks why using two source languages helps and what happens when we further increase the number of source languages. One simple explanation for the accuracy improvement is that training on two source languages has a regularization effect and prevents the model from overfitting to a particular language. Unfortunately, our dataset has a limited number of languages, and we could not experiment with more (related) source languages. Please let us know if there are other text classification datasets with more languages.
>
> In our updated draft, we try to clarify reviewer 2’s questions to provide more explanations for experiment results. Please let us know if you have any additional questions or suggestions.

---

### Official Review · AnonReviewer3 · 2018-11-03
**Straightforward model, sub-par experiment setup**

**Rating:** 4
**Confidence:** 3

**Review:**

The paper proposes to transfer document classifiers between (closely) related languages by exploiting cross-lingual subword representations in a cross-lingual embedder jointly with word-based classifier: the embedder represents the words, while the classifier labels the document. The approach is reasonable, albeit somewhat unexciting, as the basic underlying ideas are in the vein of Pinter et al. (2017), even if applied on a different task.

The main concern I have with the paper is that it leaves much open in terms of exploring the dimension of (dis)similarity: How does the model perform when similarity decreases across language pairs in the transfer? The paper currently offers a rather biased view: the couplings French-Italian-Spanish, Danish-Swedish are all very closely related languages, and Amharic-Tigrinya are also significantly related. Outside these couplings, there's a paragraph to note that the method breaks down (Table 5 in the appendix). Sharing between Romance and Germanic languages is far from representative of "loosely related languages", for all the cross-cultural influences that the two groups share.

While the experiment is reasonably posed, in my view it lacks the cross-lingual breadth and an empirical account of similarity. What we do in cross-lingual processing is: port models from resource-rich to low-resource languages, and to port between very similar languages that already have resources is a purely academic exercise. This is not to say that evaluation by proxy should be banned, but rather that low-resource setups should be more extensively controlled for.

Thus, in summary, a rather straightforward contribution to computational modeling paired with sub-par experiment setup in my view amounts to a rejection. The paper can be improved by extending the experiment and controlling for similarity, rather than leaving it as implication.

---

> ### Author Response · Authors · 2018-11-26
> **Response**
>
> Thank you for your review!
>
> Reviewer 3 comments that our method are “unexciting” and similar to Pinter et al. (2017). We believe the novelty lies in our application to cross-lingual document classification and our multi-task objective. The objective proposed by Pinter et al. (2017) is only one of the three auxiliary tasks, and they only apply their model to monolingual tasks.
>
> Reviewer 3’s main concern is that our experiments do not cover enough language pairs (with different amount of similarities). We have made our best effort to cover a diverse set of language pairs with. While we wish to investigate more language pairs, we cannot find a dataset for further experiments. RCV2 is a standard benchmark of cross-lingual document classification, and yet most of the languages (with enough labeled documents) are Indo-European. Please let us know if there are other text classification datasets with more languages.
>
> Transferring between similar languages is more than an “academic exercise”. Sometimes related languages have very different amount of resources. For example, Hindi has almost ten times as many speakers as Urdu. In our experiments, we construct datasets for truly low-resource language such as Tigrinya and experimentally demonstrate the effectiveness of our method.

---

### Author Response · Authors · 2018-11-26
**Updates**

We sincerely thank all reviewers for the useful reviews. We have uploaded a revised paper to address some of the questions and suggestions. We experiment with two additional models:
1. A lightly-supervised monolingual model trained on fifty labeled target language document (SUP in Table 1).
2. A combined model that adds CLWE as additional features for the CACO classifier (COM in Table 1). This model achieves a significantly higher average test accuracy, which shows our model is useful even when we have enough resources to train a good CLWE.

Please see our responses for detailed discussion.

---

### Meta-Review · Area_Chair1 · 2018-12-14

**Confidence:** 4
**Recommendation:** Reject

**Metareview:**

The paper's contribution lies in using cross-lingual sharing of subword representations for improving document classification.  The paper presents interesting models and results.

While the paper is good (two out of three reviewers are happy about it), I do agree with the reviewer who suggests the experimentation with relatively dissimilar languages and showing whether or not the approach works for those cases.  I am also not very happy with the author response to the reviewer.  Moreover, I think the paper could improve further if the authors presented experiments on more tasks apart from document classification.